# Data Attribution for Diffusion Models: Timestep-induced Bias in Influence Estimation

**Tong Xie**[*1]                                               *tongxie@ucla.edu*
**Haoyu Li**[*1]                                               *haoyuli02@ucla.edu*
**Andrew Bai**[2]                                              *andrewbai@cs.ucla.edu*
**Cho-Jui Hsieh**[2]                                           *chohsieh@cs.ucla.edu*
[1]*Department of Mathematics,* [2]*Department of Computer Science*
*University of California, Los Angeles*

**Reviewed on OpenReview:** *https://openreview.net/forum?id=P3Lyun7CZs*

## Abstract

Data attribution methods trace model behavior back to its training dataset, offering an effective approach to better understand "black-box" neural networks. While prior research established quantifiable links between model output and training data in diverse settings, interpreting diffusion model outputs in relation to training samples remains underexplored. In particular, diffusion models operate over a sequence of timesteps instead of instantaneous input-output relationships in previous contexts, posing a significant challenge to extend existing frameworks to diffusion models directly. Notably, we present Diffusion-TracIn that incorporates this temporal dynamics and observe that samples' loss gradient norms are highly dependent on timestep. This trend leads to a prominent bias in influence estimation, and is particularly severe for samples trained on large-norm-inducing timesteps, causing them to be generally influential. To mitigate this bias, we introduce Diffusion-ReTrac as a re-normalized adaptation that retrieves training samples targeted to the test sample of interest, enabling a localized measurement of influence and considerably more intuitive visualization. We demonstrate the efficacy of our approach through various evaluation metrics and auxiliary tasks, outperforming in terms of specificity of attribution by over 60%.

## 1 Introduction

Deep neural networks have emerged to be powerful tools for the modeling of complex data distributions and intricate representation learning. However, their astounding performance often comes at the cost of interpretability, leading to an increasing research interest to better explain these "black-box" methods. Instance-based interpretation is one approach to explain why a given machine learning model makes certain predictions by tracing the output back to training samples. While these methods have been widely studied in supervised tasks and demonstrated good performance (Koh & Liang, 2017; Yeh et al., 2018; Pruthi et al., 2020), there is limited exploration of their application in unsupervised settings, especially for generative models (Kingma & Welling, 2013; Goodfellow et al., 2020; Ho et al., 2020). In particular, diffusion models represent a state-of-the-art advancement in generative models and demonstrate remarkable performance in a variety of applications such as image generation, audio synthesis, and video generation (Kong et al., 2020; Dhariwal & Nichol, 2021; Ho & Salimans, 2022; Saharia et al., 2022a; Hertz et al., 2022; Li et al., 2022; Ho et al., 2022). The prevailing generative agents in creative arts also call for fair attribution methods to acknowledge the training data contributors (Rombach et al., 2022; Ramesh et al., 2022; Saharia et al., 2022b; Zhang et al., 2023; Brooks et al., 2023). Nonetheless, the interpretability and attribution of diffusion models remain an under-explored area (Georgiev et al., 2023; Dai & Gifford, 2023).

---

[*]Equal contribution

Compared to traditional supervised settings, the direct extension of instance-based interpretation to diffusion models is challenging due to the following factors. First, the diffusion objective involves an expectation over the injected noise $\epsilon \sim \mathcal{N}(0, I)$, hence a precise computation is impractical. Second, diffusion models operate over a sequence of timesteps instead of instantaneous input-output relationships. Although each timestep is weighted equally during the training process, we observe that certain timesteps can exhibit high gradient norm. This means the gradient of the diffusion loss function with respect to model parameters is dominantly large relative to all other timesteps (Figure 2). As most instance-based explanation models utilize this first-order gradient information, such biased gradient norms can propagate its domination into the influence estimation for diffusion models. In practice, timesteps are often randomly sampled during training. This introduces arbitrariness in influence calculation, since a training sample that happens to be trained on certain timesteps may exhibit higher-than-usual gradient norms, and thus be characterized as "generally influential" to various completely different test samples.

We present Diffusion-TracIn and Diffusion-ReTrac to demonstrate and address the existing difficulties. Diffusion-TracIn is a designed extension of TracIn (Pruthi et al., 2020) to diffusion models that incorporates the denoising timestep trajectory. This approach showcases instances where influence estimation is biased. Subsequently, we introduce Diffusion-ReTrac as a re-normalization of Diffusion-TracIn to alleviate the dominating norm effect.

Our contributions are summarized as follows:

1. Propose Diffusion-TracIn as a designed extension to diffusion models that incorporates and effectively approximates the timestep dynamics.

2. Identify and investigate the timestep-induced gradient norm bias in diffusion models, providing preliminary insights into its impact on influence estimation.

3. Introduce Diffusion-ReTrac to mitigate the timestep-induced bias, offering targeted data attribution.

4. Illustrate and compare the effectiveness of the proposed approach to address the gradient norm bias on auxiliary tasks.

## 2 Related Work

Data attribution methods trace model interpretability back to the training dataset, aiming to answer the following fundamental question: *which training samples are most responsible for shaping model behavior?*

This proves valuable across a wide range of domains, such as outlier detection, data cleaning, data curating, and memorization analysis (Khanna et al., 2019; Liu et al., 2021; Kong et al., 2021; Lin et al., 2022; Feldman, 2020; van den Burg & Williams, 2021). The adoption of diffusion models in artistic pursuits, such as Stable Diffusion and its variants, has also gained substantial influence (Rombach et al., 2022; Zhang et al., 2023). This then calls for fair attribution methods to acknowledge and credit artists whose works have shaped these models' training. Such methods are also vital for addressing associated legal and privacy concerns (Carlini et al., 2023; Somepalli et al., 2023).

### 2.1 Influence Estimations

Influence functions quantify the importance of a training sample by estimating the effect induced when the sample of interest is removed from training (Koh & Liang, 2017). This method involves inverting the Hessian of loss, which is computationally intensive and can be fragile in highly non-convex deep neural networks (Basu et al., 2020). Representer Point is another technique that computes influence using the representer theorem, yet relies on the assumption that attribution can be approximated by the final layer of neural networks, which may not hold in practice (Yeh et al., 2018). For diffusion models, the application of influence functions is significantly hindered by its computational expense while extending the representer point method is ambiguous due to the lack of a natural "final layer." Pruthi et al. (2020) introduced TracIn to measure influence based on first-order gradient approximation that does not rely on optimality conditions.

Recently, TRAK is introduced as an attribution method for large-scale models, which requires a designed ensemble of models and hence is less suitable for naturally trained models (Park et al., 2023).

In this paper, we extend the TracIn framework to propose an instance-based interpretation method specific to the diffusion model architecture. To address the challenges arose, we present Diffusion-ReTrac that re-normalizes the gradient information and effectively accommodates the timesteps dynamics. Previous works have utilized similar re-normalization techniques to enhance influence estimator performance in supervised settings. RelatIF reweights influence function estimations using optimization objectives that place constraints on global influence, enabling the retrieval of explanatory examples more localized to model predictions (Barshan et al., 2020). Gradient aggregated similarity (GAS) leverages re-normalization to better identify adversarial instances (Hammoudeh & Lowd, 2022). These works align well with our studies in understanding the localized impact of training instances on model behavior.

## 2.2 Influence in Unsupervised Settings

The aforementioned methods address the fundamental question in supervised settings, where model behavior may be characterized in terms of model prediction and accuracy. However, extending this framework to unsupervised settings is non-trivial due to the lack of labels or ground truth. It is challenging to quantify the influence of specific training samples on model behavior, as there lacks a straightforward metric to gauge the change in accuracy or relevance of the output.

Prior works explore this topic and approach to compute influence for Generative Adversarial Networks (GAN) (Terashita et al., 2021) and Variational Autoencoders (VAE) (Kong & Chaudhuri, 2021). Previous work in diffusion models quantifies influence through the use of ensembles, which requires training multiple models with subsets of the training dataset, making it unsuitable for naturally trained models (Dai & Gifford, 2023). Journey-TRAK applies TRAK (Park et al., 2023) to diffusion models and attributes each denoising timestep individually (Georgiev et al., 2023). As the diffusion trajectory spans numerous timesteps, this approach results in a multitude of attribution scores for a single image, posing challenges to settings where a holistic single-shot attribution is more interpretable and desired. Additionally, D-TRAK presents counter-intuitive findings suggesting that theoretically unjustified design choices in TRAK result in improved performance, emphasizing the need for further exploration in data attribution for diffusion models (Zheng et al., 2023). These works are complementary to our studies and contribute to a more comprehensive understanding of instance-based interpretations in unsupervised settings.

## 3 Preliminaries

### 3.1 Diffusion Models

Denoising Diffusion Probabilistic Models (DDPMs) (Ho et al., 2020) are a special type of generative models that parameterized the data as $p_\theta(x_0) = \int p_\theta(x_{0:T}) \, dx_{1:T}$, where $x_1, \ldots, x_T$ are latent variables of the same dimension as the input. The inner term $p_\theta(x_{0:T})$ is the reverse process starting at the standard normal $p(x_T) = \mathcal{N}(x_T; 0, I)$, which is defined by a Markov chain:

$$p_\theta(x_{0:T}) = p(x_T) \prod_{t=1}^{T} p_\theta(x_{t-1}|x_t), \tag{1}$$

where $p(x_{t-1}|x_t) = \mathcal{N}(x_{t-1}; \mu_\theta(x_t, t), \Sigma_\theta(x_t, t))$. The reverse process is learnable and fixed to a Markov Chain based on a variance scheduler $\beta_1, \ldots, \beta_T$. Notably, efficient training is achieved by stochastically selecting timesteps for each sample. This means samples are trained on different timesteps in an epoch, allowing for coverage of the trajectory over time without the need to process each timestep sequentially. DDPM further simplifies the loss by re-weighting each timestep, leading to the training objective used in practice,

$$L_{\text{simple}}(\theta) = \mathbb{E}_{x_0, t, \epsilon}[d(\epsilon, \epsilon_\theta(\sqrt{\alpha_t}x_0 + \sqrt{(1 - \alpha_t)}\epsilon, t)], \tag{2}$$

where $\epsilon \sim \mathcal{N}(0, I)$ and $d$ is the loss function such as $l_1$ or $l_2$ distance.

### 3.2 TracIn

TracIn is proposed as an efficient first-order approximation of a training sample's influence. It defines the idealized version of influence of a training sample $z$ to a test sample $z'$ as the total reduction of loss on $z'$ when the model is trained on $z$. For tractable computation, this change in loss on the test sample is approximated by the following Taylor expansion centered at the model parameters $w_k$

$$\ell(w_{k+1}, z') - \ell(w_k, z') = \nabla\ell(w_k, z') \cdot (w_{k+1} - w_k) + \mathcal{O}(\|w_{k+1} - w_k\|^2). \tag{3}$$

If stochastic gradient descent (SGD) is utilized in training, then the parameter update is measured by $w_{k+1} - w_k = -\eta_t \nabla\ell(w_k, z_k)$. Therefore, the first-order approximation of Ideal-Influence is derived to be

$$\text{TracIn}(z, z') = \sum_{k:z_k=z} \eta_k \nabla\ell(w_k, z') \cdot \nabla\ell(w_k, z), \tag{4}$$

summing over iterations $k$ where the particular training sample $z$ is utilized. To reduce computational costs, TracIn approximates the influence with saved checkpoints from training to replay the entire training process. Typically, 3 to 20 checkpoints with a steady decrease in loss are utilized, reducing the excessive storage of checkpoints while effectively capturing training. It is also expected that the practical form of TracIn remains the same across variations in training, such as optimizers, learning rate schedules, and handling of minibatches (Pruthi et al., 2020).

## 4 Extending TracIn to Diffusion Models & Challenges

### 4.1 Diffusion-TracIn

In this section, we present Diffusion-TracIn to provide an efficient extension of TracIn designed for diffusion models. Specifically, two adjustments keen to diffusion models are critical to enable this extension. First, the diffusion objective is an expectation of denoising losses over different timesteps $1 \leq t \leq T$. Second, the objective involves an expectation over the added noise $\epsilon \sim \mathcal{N}(0, I)$. To address these challenges, we first apply TracIn conditioned on each timestep $t$, then we compute a Monte Carlo average over $m$ randomly sampled noises $\epsilon$.

From equation 2, it is possible to treat the diffusion model learning objective as a combination of $T$ loss functions. If we denote $L_t(\theta, \epsilon, x_0) = L_{\text{simple}}(\theta, \epsilon, x_0, t) := d(\epsilon, \epsilon_\theta(\sqrt{\alpha_t}x_0 + \sqrt{(1 - \alpha_t)}\epsilon, t)$ to be a distinct loss function on each timestep $t$ and noise, we can treat $L_{\text{simple}}$ as an expectation over all the $L_t : t \in [0, T]$. Subsequently, we compute a TracIn influence score over each of the timestep $t$

$$\begin{aligned}
\text{TracIn}(z, z', t) &:= \mathbb{E}_\epsilon\Big( \sum_{k:z_k=z} \eta_k \nabla_\theta L_t(\theta_k, \epsilon, z') \cdot L \Big) \\
&\approx \frac{1}{m} \sum_{i=1}^{m} \sum_{k:z_k=z} \eta_k \nabla_\theta L_t(\theta_k, \epsilon_i, z') \cdot L,
\end{aligned} \tag{5}$$

where $L = \nabla_\theta L_{t_{\text{train}}}(\theta_k, \epsilon_{\text{train}}, z)$, $t_{\text{train}}$ is the training timesteps, and $\epsilon_{\text{train}}$ is the noise utilized when training on the sample $z$. This enables a true replay of the training dynamics. Then we define Diffusion-TracIn to be the expectation over $T$ timesteps to provide an one-shot attribution score covering the full diffusion process,

$$\begin{aligned}
\text{Diffusion-TracIn}(z, z') &:= \mathbb{E}_t(\text{TracIn}(z, z', t)) \\
&= \frac{1}{T} \sum_{t=1}^{T} \text{TracIn}(z, z', t) \\
&= \frac{1}{T} \sum_{t=1}^{T} \frac{1}{m} \sum_{i=1}^{m} \sum_{k:z_k=z} \eta_k \nabla_\theta L_t(\theta_k, \epsilon_i, z') \cdot L \\
&= \sum_{k:z_k=z} \eta_k \Big( \frac{1}{mT} \sum_{t=1}^{T} \sum_{i=1}^{m} \nabla_\theta L_t(\theta_k, \epsilon_i, z') \Big) \cdot L.
\end{aligned} \tag{6}$$

The practical form of Diffusion-TracIn also employs training checkpoints, as suggested by Pruthi et al. (2020) to enhance computational efficiency,

$$\text{Diffusion-TracIn}(z, z') := \sum_{k=1}^{s} \eta_k \left( \frac{1}{mT} \sum_{t=1}^{T} \sum_{i=1}^{m} \nabla_\theta L_t(\theta_k, \epsilon_i, z') \right) \cdot L, \tag{7}$$

where $s$ is the number of checkpoints.

### 4.2 Timestep-induced Norm Bias

To capture the influence of training samples on the model, an attribution method $A : \mathbb{R}^n \times \mathbb{R}^n \to \mathbb{R}$ assigns attribution score $a_i$ to each training sample $z_i \in D$. Therefore, an ideal and fair attribution method should derive its results based on properties intrinsic to training samples. This ensures the method's outcomes accurately capture the inherent characteristics and significance of each training sample within the dataset.

In most machine learning models, a dominating gradient norm can be largely attributed to the training sample itself. For example, outliers and samples near the decision boundary may exhibit higher gradient norms than usual. However, while sample-induced gradient norms are informative for influence estimation, we observe that the variance in gradient norms for diffusion models can also be an artifact of the diffusion training dynamics. We identify the *dominating norm effect*, which refers to phenomenon that the variance in samples' loss gradient norms being largely affected by diffusion timesteps. The timestep artifacts subsequently propagate into influence estimation, introducing 'noise' that overwhelms the genuine signal from samples, dominating and rendering the attribution results somewhat arbitrary.

Using a diffusion model trained on Artbench-2 with 10,000 samples as illustration, we demonstrate the presence of such timestep-induced norm bias by showing:

1. Statistically significant correlation exists between training samples' gradient norm and their timestep.

2. For each individual sample, its gradient norm is highly dependent on the timestep.

3. Manipulating the timestep in influence estimation significantly alters attribution results.

**Definition 1.** We define the **highest norm inducing timestep** for sample $z$ to be

$$t_{\max}(z) = \arg\max_i \|L_{t_i}(\theta, \epsilon_{\text{train}}, z)\|.$$

**Norm vs. Timestep.** We examine the distribution of training samples' loss gradient norm and the training timestep (Figure 2). This is conducted for multiple checkpoints, each yielding a distribution displaying norms at its specific stage of the training process. The distributions demonstrate a notable upward trend that peaks at, in this case, the later timestep region (i.e. closer to noise). This suggests that samples whose training timestep falls within the later range tend to exhibit higher norms.

We further examine the impact and quantify the relationship between loss gradient norms and training timesteps. Using randomly selected samples $\{z_i\}_{i=1}^{50}$, we measure the correlation between the proximity of sample $z_i$'s training timestep to $t_{\max}(z_i)$ and ranking of $z_i$'s norm among the entire training dataset. The detailed procedure is included in Algorithm 1. This is conducted for every checkpoint used to compute influence. As an illustration, Figure 1 shows a checkpoint with a correlation of 0.7 and

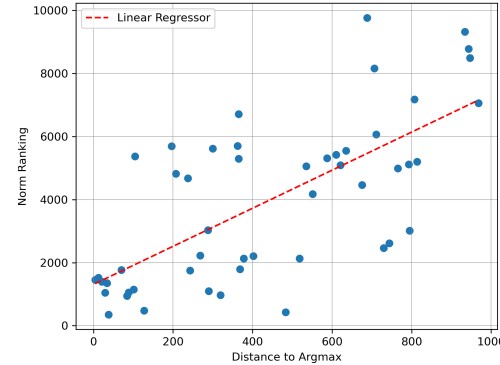

Figure 1: **Correlation of Timestep and Norm.** We plot the norm ranking and distance between training timestep to $t_{\max}$ for 50 randomly selected samples. The resulting correlation is 0.7 and the linear regressor (red) has a slope of 6.038.

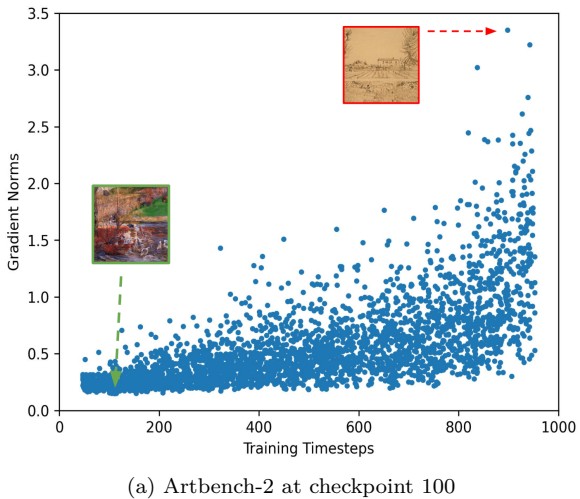
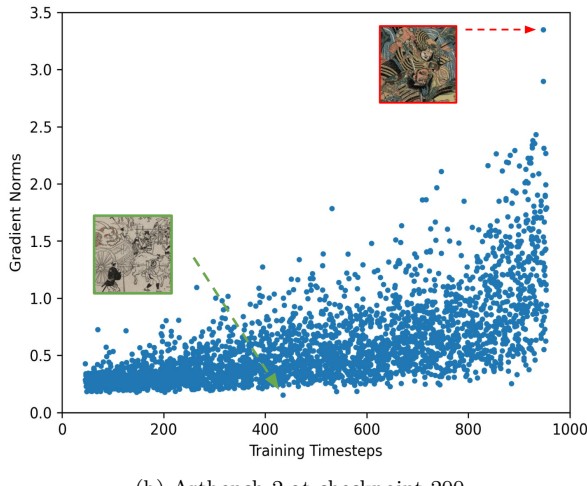

|(a) Artbench-2 at checkpoint 100 | (b) Artbench-2 at checkpoint 200|

Figure 2: **Samples' Norm vs. Training Timestep.** We plot the norm and timestep of 2,000 randomly selected training samples. We observe that loss gradient norms tend to increase when the training timestep falls in the later range (towards noise). This upward trend is consistent at other checkpoints tested. The sample with the largest norm (red) and smallest norm (green) are shown; no exceptional visual patterns are noticed. Additional distributions for others as well as open-sourced models are included in Appendix A.1 and A.2.

$p$-value $1.38 \times 10^{-7}$. A linear regressor is also fitted to the 50 data points, giving a slope of 6.038 and $p$-value $2.55 \times 10^{-8}$. This suggests a statistically significant positive correlation between the norms and training timesteps. Consequently, it indicates a notable training timestep-induced norm bias that could well dominate over sample-induced norms, which will then propagate into influence estimation.

**Varying Timestep for a Single Sample.** We further analyze the norm distribution for an individual training sample. At a given model checkpoint, we compute the loss gradient norm for a fixed sample $z$ at every timestep. We plot the norm distribution for randomly selected samples and observe similar trends. Figure 3 shows the distribution for an example training data at three different model stages. The highest norm-inducing region at these three checkpoints all falls within the later timestep range, regardless of where the training timestep is sampled at. The further implication is that for each sample, its loss gradient norm is highly dependent on the chosen timestep. It is also observed that within the same epoch, various samples share similar trends in norm distribution. This suggests a systematic pattern (e.g. artifact of diffusion learning dynamics) beyond individual instances, supporting the intuition that over-reliance on gradient norms may not be ideal.

**Timestep Manipulation.** We further illustrate the timestep-induced bias by exploring the susceptibility of influence estimation to the manipulation of timesteps. We conduct the experiment on 500 training samples that are characterized by Diffusion-TracIn as "uninfluential" to a random test sample (i.e. influence score is close to 0, neither proponent nor opponent). For each uninfluential sample $z_i$, we compute its influence using $t_{\max}(z_i)$ instead of the original training timestep. The result shows that after deliberately modifying timestep, the ranking of the magnitude of influence for these samples increases by 4,287 positions on average. Given that there are only 10,000 images in the dataset, this notable 42.8% of fluctuation indicates that the timestep-induced bias is significant enough to flip a training sample from uninfluential to proponents or opponents. It further indicates that Diffusion-TracIn's attribution results could well arise from timestep-induced norms in general. Such findings highlight the potential vulnerability within Diffusion-TracIn as a data attribution method, emphasizing the need for more robust influence estimation techniques.

The preceding sections empirically show that a training sample's loss gradient norm is highly dependent on timesteps, which produces a consistent trend in norms based on learning dynamics. As the timestep utilized in influence estimation is stochastically sampled in training, instances receive varying degrees of such timestep-induced norm. This bias is particularly evident in samples whose training timestep falls close

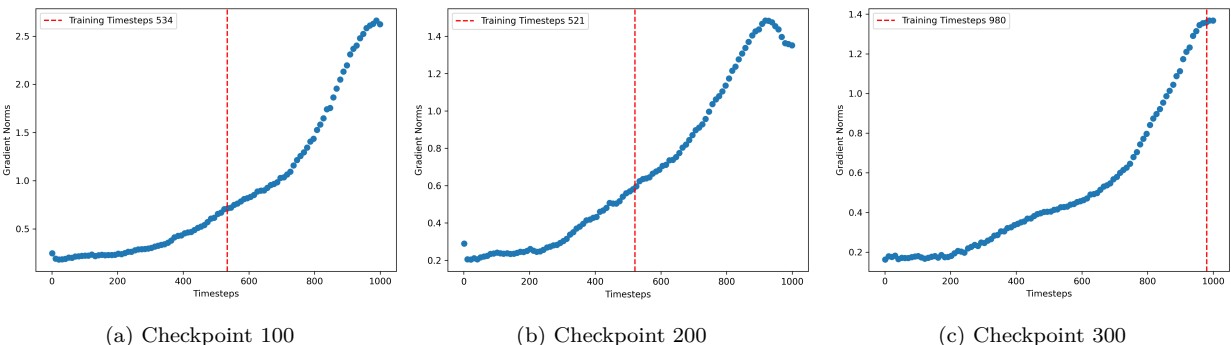

(a) Checkpoint 100           (b) Checkpoint 200           (c) Checkpoint 300

Figure 3: **One Sample's Norms Varying Timestep.** Example of norm distributions for one randomly selected sample is shown. On each checkpoint, the norm distribution is skewed to later timesteps for each individual sample. This trend exists regardless of where the actual training timestep (red) is sampled at.

to $t_{\max}$, leading them to be "generally influential" (Figure 6). These findings suggest samples' norms are a suboptimal source of information, calling for special attention to the handling of gradient norms.

## 5 Method

### 5.1 Timesteps Sparse Sampling

The extension of TracIn to diffusion models, as formulated in Equation 7, involves an expectation over $T$ timesteps to replay the training process. However, since the number of timesteps can be large in practice (e.g. 1,000), optimization toward computational efficiency becomes imperative.

To address this, we employ a reduced subset of timesteps selected to encapsulate essential information of the sequence, while retaining the diffusion learning dynamics. We leverage $n_t$ evenly-spaced timesteps $S = \{t_1, t_2, ..., t_{n_t}\}$ ranging from 1 to $T$ for the test sample $z'$ and define

$$\text{Diffusion-TracIn}(z, z') := \sum_{k=1}^{s} \eta_k \big( \frac{1}{n_t m} \sum_{t \in S} \sum_{i=1}^{m} \nabla_\theta L_t(\theta_k, \epsilon_i, z') \big) \cdot L, \tag{8}$$

where the timesteps $t_p \in S$ approximates the full diffusion trajectory.

### 5.2 Diffusion-ReTrac

As shown in Section 4.2, the direct application of TracIn to diffusion models leads to a strong dependence on timesteps and the corresponding norm induced for the training sample. The prevalence of this *dominating norm effect* calls for modification of the approach when applying to diffusion models. We thus propose Diffusion-ReTrac as a re-normalized adaptation to mitigate this effect.

**Intuition.** By Cauchy-Schwarz inequality, it can be noticed from Equation 4 that

$$|\text{TracIn}(z, z')| \leq \sum \eta_t \|\nabla \ell(w_t, z')\| \|\nabla \ell(w_t, z)\|.$$

Hence, training samples with disproportionately large gradient norms tend to have significantly higher influence score $|\text{TracIn}(z, z')|$. This suggests that these samples are more likely to be characterized as either a strong proponent or opponent to the given test sample $z'$, depending on the direction alignment of $\nabla \ell(w_t, z')$ and $\nabla \ell(w_t, z)$. As identified in Section 4.2, the loss function component $L_t$ from certain timesteps is more likely to have a larger gradient norm. Hence varying degrees of "timestep-induced" norm biases, stemming from systematic learning dynamics, emerge and affect the influence estimations.

**Approach.** Since the gradient norm is not exclusively a property of the sample, an ideal instance-based interpretation should not overestimate the influence of samples with large norms or penalize those with small norms. In fact, this dominating norm effect can be introduced by the timesteps for both test sample $z'$ and each training sample $z$, whose loss gradient norms are computed using timestep $t$ and $t_{\text{train}}$ respectively.

For test sample $z'$, the gradient information $\sum_{t \in S} \sum_{i=1}^{m} \nabla_\theta L_t(\theta_k, \epsilon_i, z')$ derives from an expectation over the $n_t$ sub-sampled timesteps $S = \{t_1, \cdots, t_{n_t}\}$. Therefore, influence estimation inherently upweights timesteps with larger norms and downweights those with smaller norms. For each training sample $z$, the timestep $t_{\text{train}}$ was stochastically sampled during the training process, hence incorporating varying degrees of timestep-induced norm bias. To this end, we propose Diffusion-ReTrac which introduces normalization that reweights the training samples to address the dominating norm effect. We normalize these two terms and define $\hat{L} = \frac{L}{||L||}$ and $\hat{L}_t(\theta, z, \epsilon) = \frac{L_t(\theta, z, \epsilon)}{||\nabla_\theta L_t(\theta, z, \epsilon)||}$. Then we define

$$\text{Diffusion-ReTrac}(z, z') = \sum_{k=1}^{s} \eta_k \left( \frac{1}{n_t m} \sum_{t \in S} \sum_{i=1}^{m} \nabla_\theta \hat{L}_t(\theta_k, z', \epsilon_i) \right) \cdot \hat{L}. \tag{9}$$

The bias introduced to influence estimation due to timestep-induced norms is thus mitigated. In this way, we minimize the vulnerability that the calculated influences are dominated by training samples with a disproportionately large gradient norm arising from stochastic training.

## 6 Experiments

To interpret model behaviors through training samples, the proposed data attribution method $A$ encompasses two natural perspectives: test-influence $A(z, z')$ and self-influence $A(z, z)$. Specifically, test-influence measures the influence of each training sample $z$ on a fixed test sample $z'$, entailing a targeted analysis on a specific test sample. On the other hand, self-influence estimates the influence of a training sample on itself, providing rich knowledge about the sample itself and gauges its relative impact on the model's learning.

We evaluate Diffusion-TracIn and ReTrac under these two objectives. The experiments provide evidence of the dominating norm bias in Diffusion-TracIn attributions, presenting instances where this effect may be unnoticed in one objective yet gives rise to substantial failure in another. We address the following questions:

1. **Image Tracing**: How effective is each method at attributing the learning source of an image to the training data through *test-influence*?

2. **Targeted Attribution**: In general datasets, how does Diffusion-ReTrac outperform Diffusion-TracIn by addressing the loss gradient norm bias?

3. **Outlier Detection**: Why might the timestep-induced bias be unnoticed in detecting outliers or atypical samples by calculating *self-influence*?

### 6.1 Image Tracing

One fundamental role of data attribution methods is to trace the model's outputs back to their origins in the training samples. This idea is also utilized for analyzing *memorization* (Feldman, 2020), a behavior where the generated sample is attributed to a few nearly identical training samples. In essence, *Image source tracing* helps pinpoint specific training samples that are responsible for a generation. Thus we evaluate our methods on the question: Given a test sample, which instances in the training dataset is the model's knowledge of the test sample derived from?

**Setup.** We begin by attributing a diffusion model trained on a combination of the entire 5,000 samples from CIFAR-10 airplane subclass and 200 samples from MNIST zero subclass (Krizhevsky et al., 2009; LeCun & Cortes, 2010). Given a test sample of MNIST zero, it is expected that the 200 MNIST samples in the training dataset serve as ground truth for the image source. Similarly, a test sample of CIFAR-plane should be attributed to the 5,000 CIFAR training samples. We thus obtain an accuracy score by measuring the correctly attributed proportion among the top-$k$ influential sample.

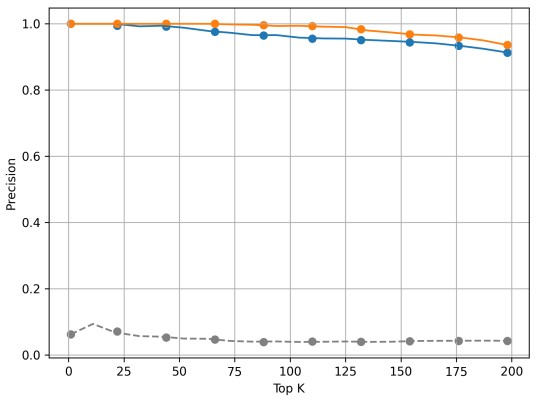
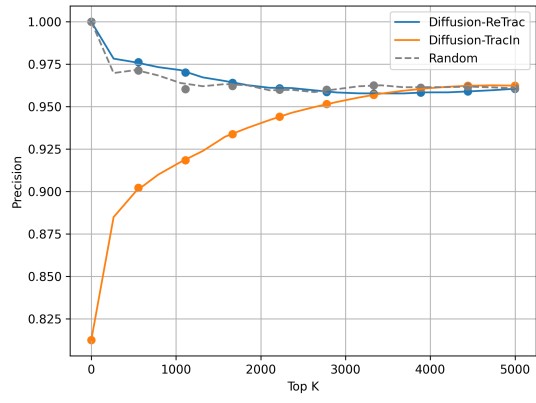

(a) Precision on MNIST zero test samples

(b) Precision on CIFAR-plane test samples

Figure 4: **Image Tracing Accuracy.** We measure precision by evaluating the ratio of correctly attributed samples among top $k$ proponents. While (a) shows both methods successfully traced MNIST test samples, (b) shows that Diffusion-TracIn fails to attribute CIFAR-plane test samples since the outlier MNIST samples with large norms are characterized as strong proponents, giving low precision for small $k$. Note that random attribution is a strong baseline in (b) since the 5,000 CIFAR-planes are more likely to be proponents compared to the 200 MNIST samples.

**Results.** The precision for Diffusion-TracIn and ReTrac are reported in Figure 4. While both methods successfully attribute the MNIST test samples to the 200 MNIST training samples, Diffusion-TracIn is significantly worse than ReTrac and the random baseline in attributing CIFAR-plane test samples. It frequently characterizes MNIST samples as top proponents for CIFAR planes. This aligns with the expectation that Diffusion-TracIn tends to assign high influence to training samples with large norms, in this case, the outlier MNIST zeros. Further analysis of these influential zeros indicates that their training timestep falls exactly at or close to $t_{max}$. This further amplifies these outliers' norms, exacerbating the bias introduced. In contrast, Diffusion-ReTrac successfully attributes both MNIST-zero and CIFAR-plane test samples. Comparison to the additional baseline of influence functions (Koh & Liang, 2017) is also included in appendix C.

**Visualization.** The attribution results are shown in Figure 5. Upon closer examining Diffusion-TracIn attribution results, the set of MNIST zero samples exerting influence on plane is relatively consistent across various CIFAR plane test samples. For instance, the MNIST sample with highest influence on the two generated planes is identical. For the CIFAR-plane samples that Diffusion-TracIn successfully attributes (without influential MNIST zeros), there still appear to be generally influential planes. This phenomenon is alleviated in ReTrac, with sets of influential samples being more distinct and visually intuitive. Additionally, the CIFAR-plane instances with high influence (e.g. among the top 200) to MNIST test samples tend to be planes with black backgrounds, which to an extent also resemble the MNIST zero. Visualization for these proponent planes is included in Appendix D.3. It is also worth highlighting that Diffusion-ReTrac identifies potentially memorized samples for the generated image, such as the last row in Figure 5.

## 6.2 Targeted Attribution

We then provide a comprehensive analysis of the influential samples retrieved by Diffusion-TracIn and ReTrac on various datasets. Compared to the previous settings, this experiment minimizes the effects of unusually large "sample-induced" gradient norms due to the deliberately introduced outliers. This experiment further compares the capability of Diffusion-TracIn and ReTrac in tasks with different emphases or objectives.

**Setup.** We compute test-influence on 4 diffusion models trained on 1). Tiny ImageNet containing 100,000 images with 64 x 64 resolution (Le & Yang, 2015), 2). CelebFaces Attributes (CelebA) subset containing 100,000 images with 128 x 128 resolution (Liu et al., 2015), 3). Artbench-2 consisting of "Post-impressionis" and "ukiyo-e" subclasses (Liao et al., 2022; Zheng et al., 2023), each with 5,000 images and resolution $64 \times 64$, and 4). CIFAR-10 (Krizhevsky et al., 2009) consisting of 50,000 images with resolution $32 \times 32$.

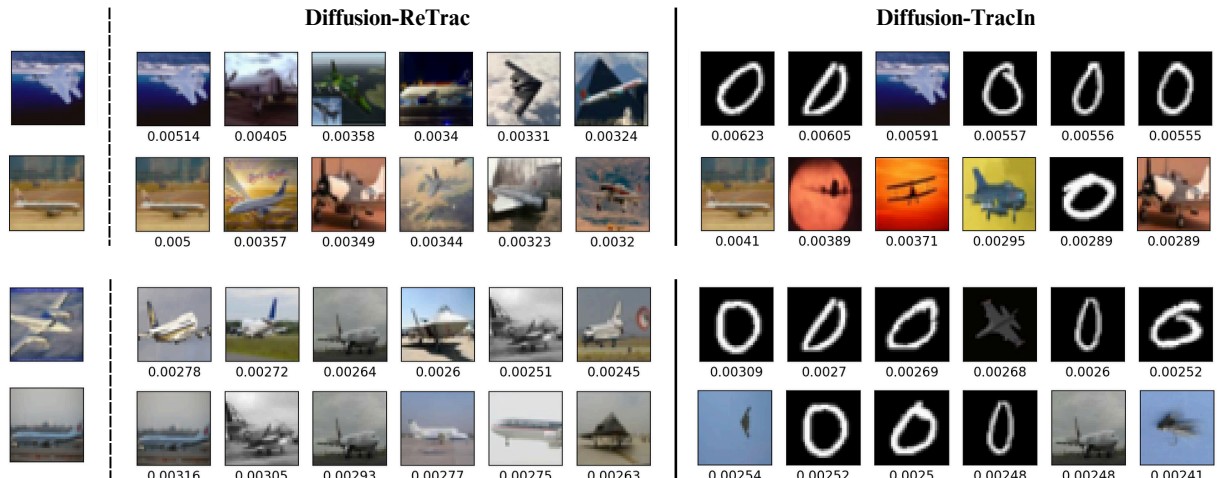

Figure 5: **Image Tracing on Outlier Model.** We show the attribution results on 4 test samples, using both training (leftmost top 2) and generated samples (leftmost bottom 2); influence scores are stated below each image. While both methods successfully attribute MNIST zero test samples, Diffusion-TracIn incorrectly characterizes MNIST training samples as influential to a CIFAR-plane test sample. This is notably mitigated with Diffusion-ReTrac.

**Results.** To quantify the targetedness of an attribution method, we assess the prevalence of generally influential samples. For a list of $n$ random test samples, we analyze the top-$k$ influential samples identified by the two methods, each giving us a total of $n \times k$ samples. Within the set, we report the proportion of distinct instances in Table 1. A lower score implies more overlapping (i.e. generally influential samples). We note that Diffusion-TracIn yields extremely homogeneous results, where the top influential training samples for a diverse set of test images are exactly the same. This is particularly evident in CelebA which is comprised entirely of human faces and ArtBench-2 containing just two subclasses, where the image categories are much less diverse and thus more susceptible to the bias induced. In these cases, Diffusion-TracIn results in only about 30% of unique samples while ReTrac retrieves up to 95.6%, significantly outperforming in terms of targetedness of attribution.

| | | Top 10 | Top 50 | Top 100 | | | Top 10 | Top 50 | Top 100 |
|---|---|---|---|---|---|---|---|---|---|
| Tiny ImageNet | D-TracIn | 0.586 | 0.474 | 0.439 | ArtBench-2 | 0.293 | 0.261 | 0.248 |
| | D-ReTrac | **0.900** | **0.841** | **0.823** | | **0.812** | **0.646** | **0.605** |
| CelebA | D-TracIn | 0.313 | 0.273 | 0.269 | CIFAR-10 | 0.725 | 0.663 | 0.636 |
| | D-ReTrac | **0.956** | **0.903** | **0.876** | | **0.856** | **0.800** | **0.768** |

Table 1: **Targeted Attribution.** This table shows the average proportion of unique samples retrieved over 16 test samples. Diffusion-TracIn extracts far fewer unique samples compared to Diffusion-ReTrac.

**Visualization.** From Figure 6, it is visually evident that Diffusion-TracIn retrieves numerous generally influential training samples. The same set of samples exhibits large influences to test samples that are completely different (e.g. in terms of subclass or visual similarities such as color and structure). In Tiny ImageNet, for instance, the training sample of a nail with white background is characterized as influential to both the red building (row 1) and the bottle (row 3), which may be visually counterintuitive.

Further analysis of these generally influential samples reveals that their associated timestep tends to be close

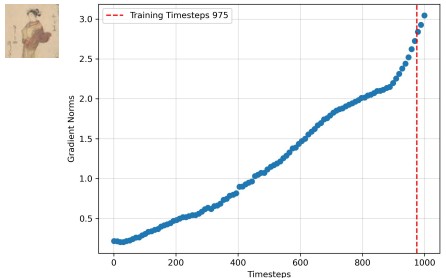

Figure 7: **Timestep and Generally Influential.** A generally influential sample whose training timestep falls exactly within $t_{\max}(x)$ region.

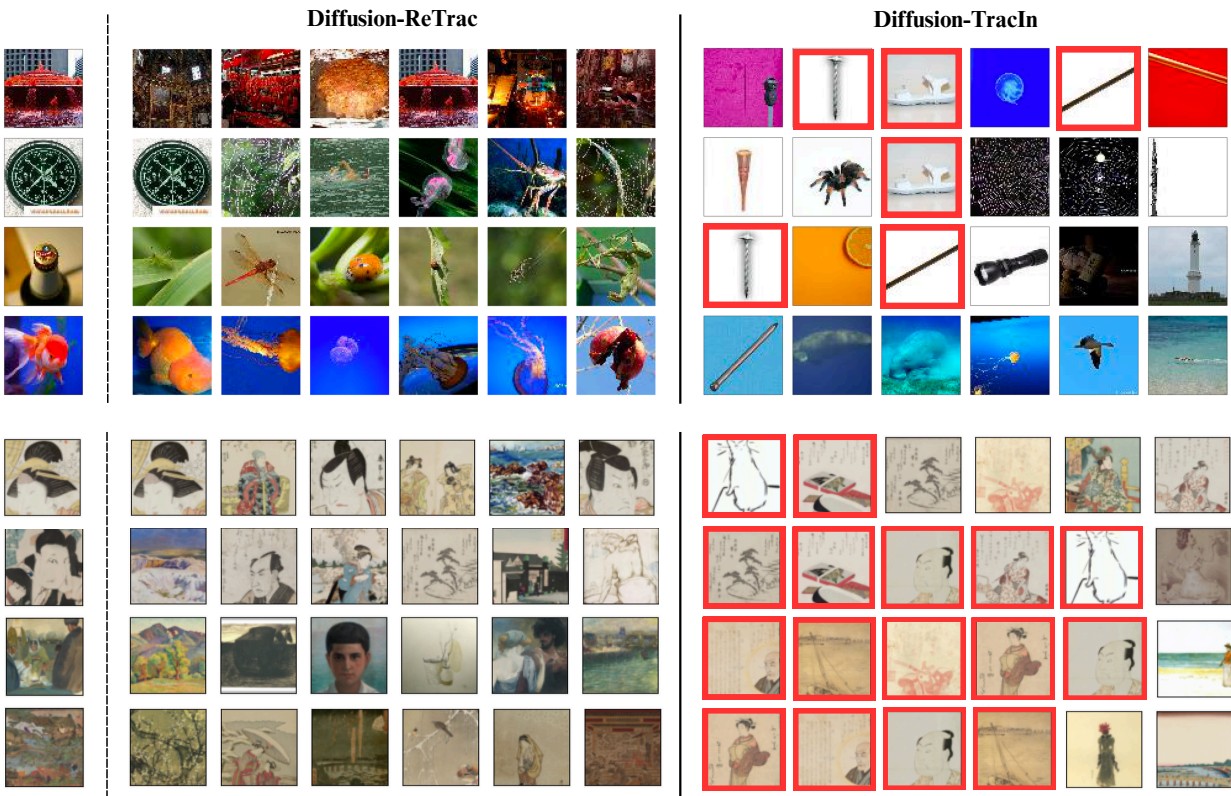

Figure 6: **Targeted Attribution.** We show attribution for 4 test samples each from Tiny ImageNet (top) and Artbench-2 (bottom). The top 6 proponents are shown. The generally influential images are indicated in red box. It is visually evident that Diffusion-ReTrac provides more distinct attribution results. More visualizations of CIFAR-10 and CelebA can be found in appendix D.1.

to $t_{\max}$. As an illustration, Figure 7 shows the distribution of norm vs. timestep for an example generally influential instance from Artbench-2. This specific sample emerges as influential at checkpoint 80, which coincides with its training timestep falling close to $T_{\max}$. It then becomes generally influential as shown in Figure 6 (first proponent on the last row). This phenomenon is notably mitigated after normalization in Diffusion-ReTrac. The revised approach retrieves proponents that bear greater visual resemblance to the test samples, highlighting ReTrac's targeted attribute and reinforcing that dissimilar test samples are more likely to be influenced by a distinct set of training samples.

## 6.3 Outlier Detection

Influence estimation is often used to identify outliers that deviate notably from the rest of the training data. Intuitively, outliers independently support the model's learning at those sparser regions of the input space to which they belong, whereas learning of the typical samples is supported by a wide range of data. Hence in an ideal influence method, outliers tend to exhibit high self-influence, indicating that they exert a high contribution in reducing their own loss. Because of such an outlier-induced norm, we observe that biased estimations may easily go unnoticed in this common metric of outlier detection.

**Setup.** We extend the experiment conducted on the CIFAR-MNIST dataset mentioned in section 6.1. We compute the self-influence of each of the training instances, and sort them by descending order. Since the 200 MNIST samples are outliers that independently support a region, we evaluate whether our methods assign high self-influence to the 200 samples of MNIST zero.

**Result.** The results show that even after normalization, ReTrac retains the ability to identify outliers. This further supports that certain variations in loss scale between timesteps are arbitrary and do not convey relevant signals (Table 2). Nonetheless, the bias introduced by diffusion timesteps is unnoticed in this experiment. Since outliers naturally exhibit larger norms compared to the typical inliers, the timestep-induced norm becomes a more obscure confounding factor and hence is less subtle in the computation of self-influence.

|  | Top 100 | Top 200 | Top 300 |
|---|---|---|---|
| Diffusion-TracIn | 0.880 | 0.880 | 1.000 |
| Diffusion-ReTrac | 0.860 | 0.845 | 1.000 |

Table 2: **Outlier Detection.** This table measures the proportion of MNIST samples among top-$k$ identified samples with the highest self-influence. The performance of these two methods is comparable. Both methods assign high self-influence to the 200 MNIST outliers out of the 5,000 CIFAR planes.

**Visualization.** Examining high-ranking samples shows that the airplane samples with high self-influence (among the top 200) have large contrast and atypical backgrounds compared to airplane samples with low self-influence. Overall, instances with high self-influence tend to exhibit high visual contrast or are difficult to recognize. This observation is consistent with patterns revealed in previous work on influence estimation for VAE (Kong & Chaudhuri, 2021). Visualization for plane samples with high self-influence is included in Appendix D.2.

## 7 Conclusion

In this work, we extend the data attribution framework to diffusion models and identify a prominent bias in influence estimation originating from loss gradient norms. Our detailed analysis elucidates how this bias propagates into the attribution process, revealing that gradient information harbors undesired bias caused by diffusion model dynamics. Subsequent experiments validate Diffusion-ReTrac as an effective attempt to mitigate this effect, offering fairer and targeted attribution results.

**Limitations and future work.** Since our proposed methods estimate influence by replaying the training process, adherence to the theoretical framework necessitates the usage of timesteps utilized during training, potentially limiting the scope of attribution. In addition, a theoretical explanation of the large-norm-inducing timesteps would offer greater insights into the underlying causes and facilitate tailored solutions. While renormalization effectively addresses the dominating norm effect and "generally influential" samples, exploring more refined normalizing strategies and further examining the gradient alignments may also be beneficial. Appendix B provides an overview of the efficacy and trade-offs of various normalization methods.

Furthermore, analysis of other potential confounding factors would provide a more comprehensive understanding of what constitutes a fair attribution method. Exploring attribution methods in latent diffusion models (Rombach et al., 2022) and scenarios involving corrupted data such as Ambient Diffusion (Daras et al., 2024) presents an intriguing avenue for future research, as it paves the way for understanding in more complex settings and the development of more interpretable methods for a wide range of applications.

## 8 Acknowledgements

This work is partially supported by NSF 2048280, 2331966, 2325121, 2244760 and ONR N00014-23-1-2300.

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

# Appendix

## A   Timestep-Induced Bias

### A.1   Norm vs. Timestep

To demonstrate that diffusion timesteps have a significant impact on loss gradient norms, we plot the distribution of 2,000 randomly selected training samples' norms and their training timesteps. Visualization for the distribution is shown in Figure 8. There is a notable upward trend that peaks at the later range of the timesteps (i.e. timesteps closer to noise), suggesting that samples trained during these later timesteps tend to exhibit larger norms. Additionally, it is also observed that the trend in norm distribution gradually diminishes at the model convergence. This further supports that such variance due to timestep is an artifact of the training dynamic, rather than a property of the training sample. However, Diffusion-TracIn utilizes gradient information throughout the entire learning process instead of focusing solely on those near convergence. This approach is due to the tendency of the latter to contain minimal information, resulting in an inevitable trend in norms affecting influence estimation. This also motivates the renormalization technique in Diffusion-ReTrac.

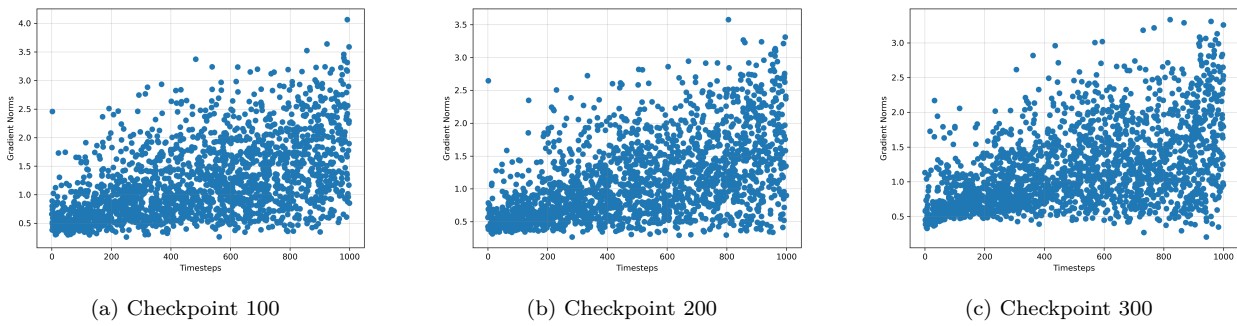

(a) Checkpoint 100        (b) Checkpoint 200        (c) Checkpoint 300

Figure 8: **Norm Distribution.** We plot the loss gradient norm and training timestep of 2,000 samples. The distributions at checkpoints 100, 200, and 300 all demonstrate an upward trend. This suggests that samples whose training timesteps fall within the later timestep region tend to have a larger norm.

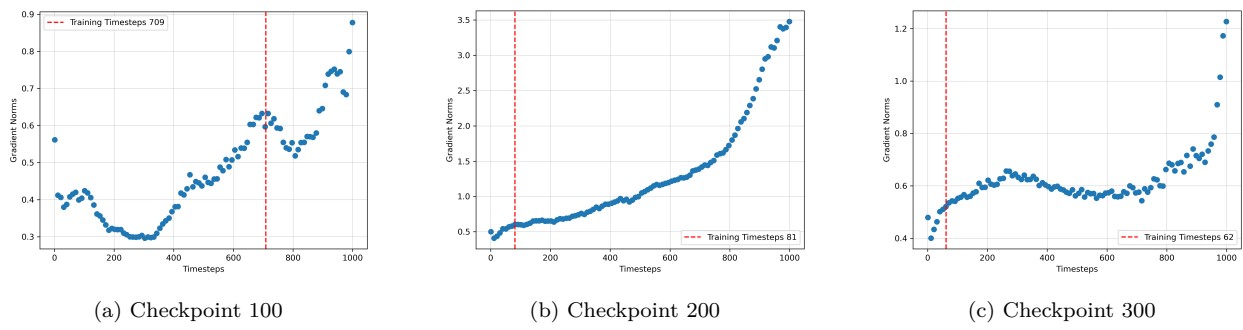

(a) Checkpoint 100        (b) Checkpoint 200        (c) Checkpoint 300

Figure 9: **Varying Timestep for a Single Sample.** The norms of sample #0 are computed at different timesteps. The distribution obtained at checkpoints 100, 200, and 300 all demonstrates a similar trend that peaks at the later timestep region.

## A.2 Norm Distributions for Various Models

To show that this observation is consistent in various diffusion models, Figure 10 plots the norm distribution on the following diffusion models: 1). trained on Tiny-ImageNet, 2). trained on CelebA, 3). open-source Google's Pretrained DDPM on CIFAR-10.

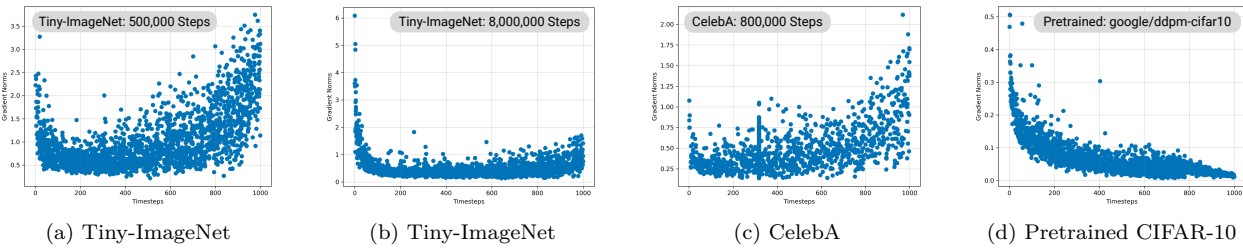

(a) Tiny-ImageNet      (b) Tiny-ImageNet      (c) CelebA      (d) Pretrained CIFAR-10

Figure 10: **Norm Distribution on Additional Models.** We plot the loss gradient norm and timestep for three additional models. There is a notable region of timesteps with significantly larger norms, although the region may differ depending on model learning dynamics.

*Why is there a correlation between norms and timesteps, where is it coming from?*

Our hypothesis regarding this trend is primarily rooted in the learning dynamics inherent to diffusion models. Specifically, previous works have suggested that later timesteps in diffusion models tend to learn structure, while earlier timesteps focus on capturing finer details Fang et al. (2024). We speculate that this observed correlation between gradient norms and training timesteps may stem from similar dynamics, as evidenced by the variability in regions of high-norm induction across different stages of learning. Our work identifies and presents an initial investigation into this issue, further theoretical exploration would be beneficial.

## A.3 Varying Timestep for a Single Sample

We further show that for a fixed training sample, the gradient norm with respect to its loss computed at different timesteps varies significantly. This reinforces the effect of training timestep in the estimation of influence, indicating that each sample receives a varying degree of bias since the training timestep is stochastically sampled. An example norm distribution for a fixed sample at different checkpoints is shown in Figure 9.

## A.4 Correlation

We provided quantitative analysis addressing the question: If the training timestep of a sample $x$ falls closer to $t_{\max}(x)$, does $x$ also have a relatively larger norm compared to the rest of the training dataset? To analyze the relationship between the stochastically chosen training timestep and the sample's overall norm ranking among the rest, we obtain a correlation score by i). compute the distance between a sample's training timesteps $t_{\text{train}}$ and the timestep that yields the maximum norm $t_{\max}(x)$, ii). the ranking of this sample's gradient norm among all the training samples, and iii). calculate a Spearman Rank correlation score between distance and ranking (Algorithm 1). Figure1 in the main text shows a visualization of the measured correlation.

---

**Algorithm 1** Training Timesteps and Norm Ranking

---

$y_1 \leftarrow \{\}$
$y_2 \leftarrow \{\}$
**for** $k \leftarrow 0$ to $N$ **do**:
    $x \leftarrow k$-th training sample
    $y_1 \leftarrow y_1 \cup |t_{\max}(x) - t_{\text{train}}(x)|$
    $y_2 \leftarrow y_2 \cup \text{Rank}(x)$
**end for**
**return** spearman-rank$(y_1, y_2)$

---

## B  Intensity of Normalization

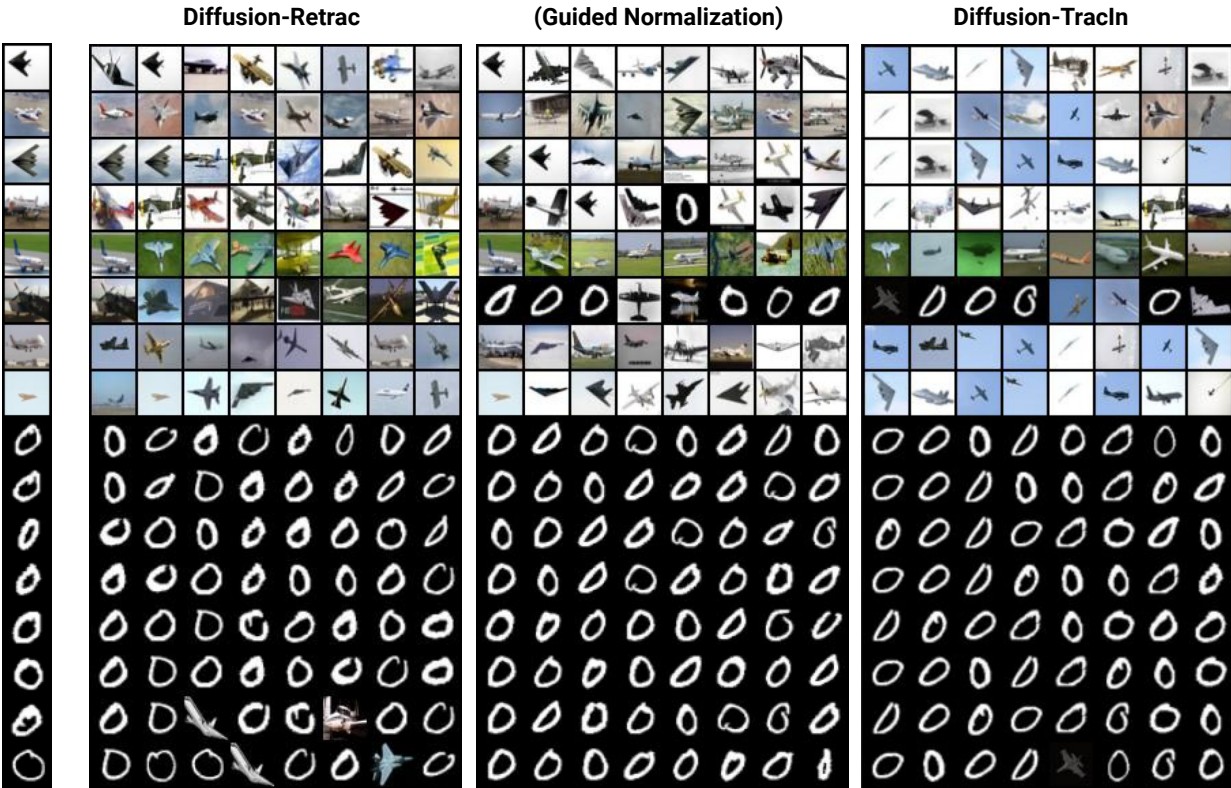

Figure 11: **Comparison of Normalization Intensity.** This shows the attribution results and associated trade-offs from three distinct approaches (non-differentiating, guided, and no normalization) for 16 test samples on the leftmost.

*What are the consequences of suppressing all norms through re-normalization techniques?*

We note that the normalization technique in Diffusion-Retrac may involve an inherent trade-off, as it diminishes the significance of certain critical timesteps by treating all norms equally. In an attempt to address this issue, we here explore the possibility of utilizing "*Guided Normalization*".

**Guided Normalization.** As our experiments suggest, the norm of a training sample is comprised of two major components: (1). natural norm that is fully dependent on the training sample's properties, and (2). dynamic norm that is highly dependent on timestep; and norm 2 is what is considered to be biasing the attribution. Guided Normalization is an attempt to only remove such timestep-induced dynamic norms, while maintaining the natural norm informative for attribution.

We fix a timestep $t$ (e.g. the highest norm-inducing timestep at that checkpoint), and compute the norm for every training sample at $t$. Since the effect of timestep is now controlled for all samples, this resulting variation in norms is sample-induced. Hence we utilize this variance between samples to guide us in answering the question *how much to normalize for each sample?* One potential approach is to set a lower threshold $\lambda$. And the training sample with the highest norm ($\text{norm}_{t_{\max}}$) remains unchanged, while subsequent samples are gradually normalized in proportion to their respective norms. This normalization process continues until the sample with the lowest norm is adjusted to ($\lambda \times \text{norm}_{t_{\max}}$). Guided Normalization is a transition in between ReTrac and TracIn, aiming at alleviating the dominant influence in Diffusion-TracIn, while retaining a precise portion of the loss gradient norm according to each sample.

**Improvements and Trade-off.** The comparison of attribution results shown in Figure 11 demonstrates this perspective of transition, where the proponents become increasingly repetitive in methods from left to right (e.g. generally influential proponent in rows 2-4 for Diffusion-TracIn). And the outliers MNIST-zero samples with larger norms are characterized as influential to CIFAR-planes. Similarly, the trade-off

associated with non-differentiating normalization in ReTrac is also evident in the attribution of MNIST-zeros, where visually similar CIFAR-planes are assigned as top proponents. It is also worth noting that in Diffusion-TracIn, if a training sample $z$ has a timestep that falls in a large-norm inducing region (such as timesteps close to 1000 in Figure 2), then this sample $z$ often emerges as one of the top proponents for most test samples in the calculation. We observe that this could be the case even when $z$ only has a large-norm timestep in one checkpoint, which means that this effect is strong enough to dominate the attribution results from previous checkpoints.

Overall, since we did not observe an immensely convincing improvement with guided normalization, we ultimately opted for the ReTrac normalization approach as a simple mitigation for the overly dominant influence that the vanilla Diffusion-TracIn exhibits. However, further refinement of Guided Normalization and other strategies for optimal normalization may be an interesting direction to explore.

## C  Comparison with Influence Functions

We evaluate the performance of Diffusion-ReTrac and Diffusion-TracIn against the Influence Functions (Koh & Liang, 2017), one of the best-known influence estimators. It considers how a model changes if training instance $z_i$'s weight is infinitesimally perturbed by $\epsilon_i$.

$$\mathcal{I}(z, z_{\text{test}}) = -\nabla_\theta L(z_{\text{test}}, \hat{\theta})^T H_{\hat{\theta}}^{-1} \nabla_\theta L(z, \hat{\theta}) \tag{10}$$

More intuitively, this approach estimates the leave-one-out influence of $z_i$ on the test sample. However, directly applying influence functions is computationally intensive as it requires inverting the Hessian, and is particularly impractical due to the sheer number of parameters in diffusion models. To overcome this limitation, we adopted the approximation method Linear time stochastic second-order algorithm (LiSSA) proposed in the work Agarwal et al. (2017), which offers a more feasible approach for computing sample influences in large-scale diffusion models. Figure 13 shows that influence functions result in much lower precision for Image Tracing proposed in Section 6.1.

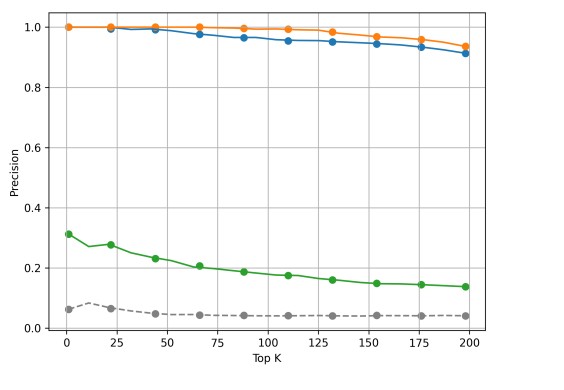
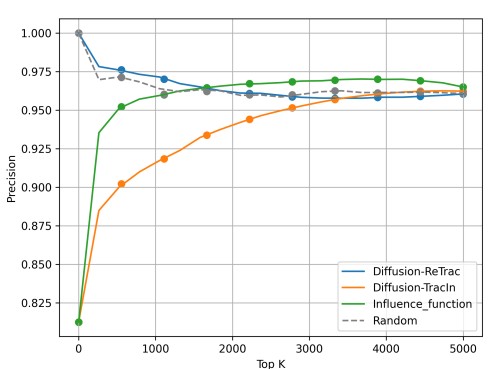

(a) Precision on MNIST zero test samples          (b) Precision on CIFAR-plane test samples

**Influence Functions**                          **Influence Functions**

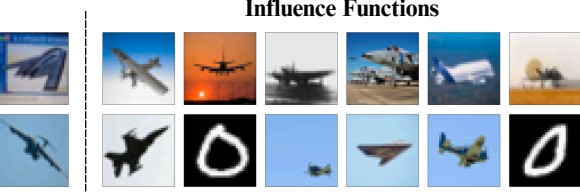
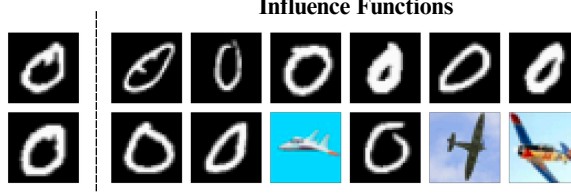

Figure 13: **Influence Functions Attribution.** This shows the top 6 proponents retrieved for the 4 test samples. There are notable cases where CIFAR-planes are attributed to MNIST-zeros and vice versa.

## D    Supplemental Visualizations

### D.1    Artbench-2 and CIFAR-10 Datasets

We provide further visualizations comparing the attribution of Diffusion-ReTrac and Diffusion-TracIn. Figure 14 shows results for 1). CelebFaces Attributes (CelebA) subset containing 100,000 images with 128 x 128 resolution (Liu et al., 2015), and 2). CIFAR-10 (Krizhevsky et al., 2009) consisting of 50,000 training samples with resolution $32 \times 32$.

Consistent with observations in the main text and data reported in Table 1, Diffusion-TracIn retrieves a set of highly homogenous training samples that are generally influential, while ReTrac significantly addresses this issue of over-dominance by certain samples. This enhancement is particularly notable in datasets like Artbench-2 and CelebA, which have less diverse training sample categories (i.e., two subclasses in Artbench-2 and exclusively human faces in CelebA). To be specific, among the top 10 influential samples for CelebA test samples, Diffusion-TracIn only results in 31.3% of unique images while ReTrac retrieves 95.6% of unique images, significantly outperforming in terms of targetedness of attribution.

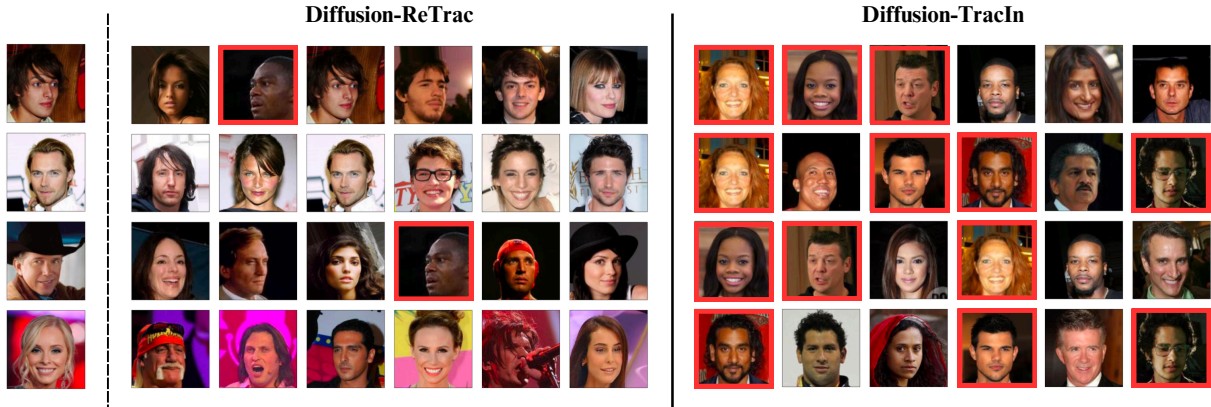

(a) **CelebA.** When the dataset is less diverse, Diffusion-TracIn retrieves many generally influential samples.

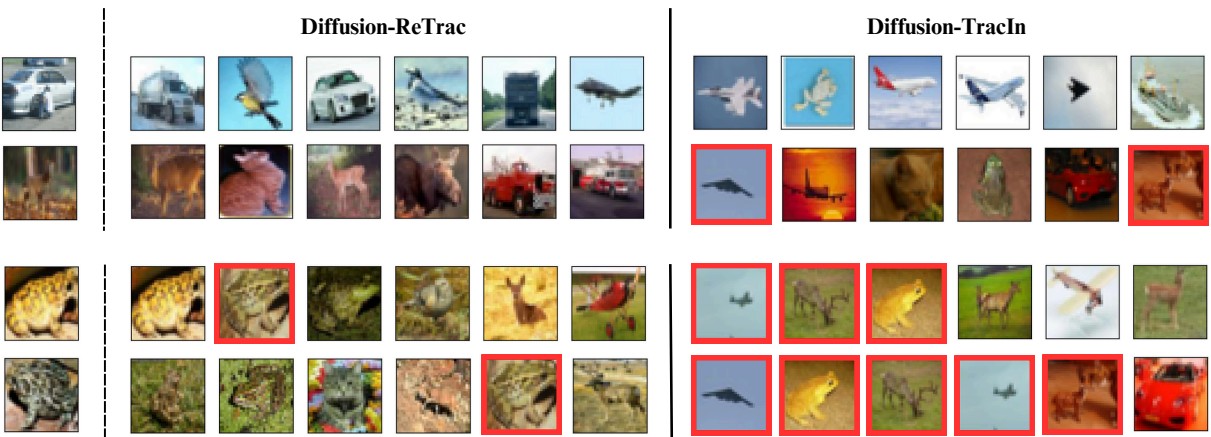

(b) **CIFAR-10.** Top 2 test samples are of the subclass "automobile," "deer" and bottom 2 are of "frog".

Figure 14: **Targeted Attribution.** We attribute the 8 test samples (leftmost) using both Diffusion-TracIn and Diffusion-ReTrac. The top 6 proponents are shown. The generally influential images that appear multiple times for different test samples are indicated in red. It is visually evident that Diffusion-ReTrac provides more distinct attribution results.

## D.2   CIFAR-Planes with High Self-influence

Self-influence is used to identify outliers in the training dataset. While Diffusion-TracIn and ReTrac assign high self-influence to most of the 200 MNIST samples, certain CIFAR-plane samples also received high self-influence scores and are ranked among the top 200. These plane samples tend to have dark backgrounds or high contrast, which are also visually distinct from typical samples in the CIFAR-plane subclass (Figure 15).

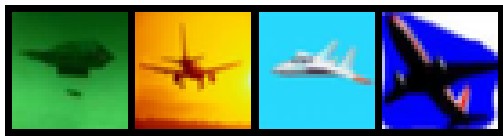

Figure 15: **CIFAR-planes with High Self-Influence.** These four samples are assigned high self-influence scores by both Diffusion-TracIn and ReTrac. They are visually distinct from the typical plane samples in the training dataset.

## D.3   CIFAR-Planes Influential to MNIST Samples

The auxiliary task of Image Source Tracing pinpoints specific training samples that are responsible for the generation of a test sample. For an MNIST zero, while most of the retrieved proponents are MNIST zeros, some planes are also assigned high influences (Figure 16). We noticed that these planes are visually distinct from the other, and visually resemble the MNIST samples. They tend to exhibit a black background and the planes are centered in the middle, which highly resembles the layout of MNIST zeros. This further proves the effectiveness of Diffusion-ReTrac in identifying highly influential samples.

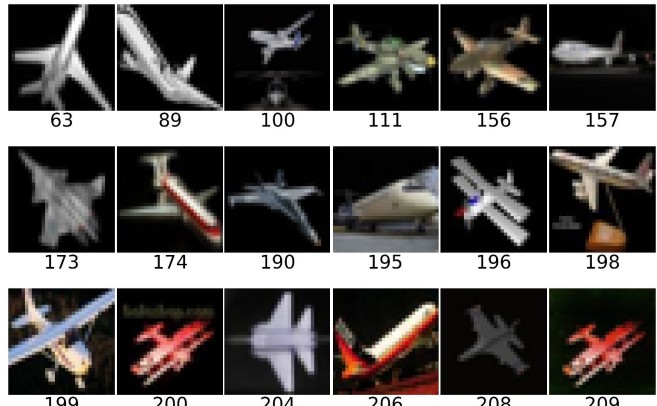

Figure 16: **CIFAR-planes Influential to MNIST Samples.** These CIFAR-Planes are assigned high influence scores to an MNIST zero test sample. The attribution results are in descending order and the corresponding ranking for each sample is labeled.

## E   Timestep Selection

### E.1   Effectiveness of Timestep Sparse Sampling

To approximate the expectation over timesteps in equation 7 efficiently, 50 linearly spaced timesteps over the denoising trajectory are used. Empirically, we demonstrate the effectiveness of this sampling approach by comparing the Diffusion-TracIn and ReTrac results obtained using our sparse sampling method against those obtained through the full expectation of all $T$ timesteps.

Specifically, we evaluate the Spearman's Rank Correlation between the attribution results derived from the two approaches. To ensure the reliability of our findings, we report the average correlation over 16 test samples. It is observed that Spearman's rank correlation consistently exceeds 0.99 across various sparsity levels (i.e., using 50/100/200 evenly spaced timesteps), for both TracIn and ReTrac. This suggests that our approach, leveraging sparse sampling, yields attribution results that closely align with those derived from full expectation. It is also worth noticing that fixing one timesteps per checkpoint is unfaithful to the true attribution result, where the correlation drops to 0.73 for Diffusion-ReTrac and 0.72 for Diffusion-TracIn. The correlation scores are organized in the table 3.

|                    | Fixed one | 50    | 100   | 200   |
| ------------------ | --------- | ----- | ----- | ----- |
| **Diffusion-TracIn** | 0.720   |       | 0.995 | 0.998 | 0.999 |
| **Diffusion-ReTrac** | 0.730   |       | 0.995 | 0.997 | 0.999 |

Table 3: **Consistency to Full Expectation.** Spearman's rank correlation between attribution results derived using sparse timesteps subsampling and full expectation over the entire diffusion trajectory.

|                    | Fixed one | 50    | 100   | 200   | Full  |
| ------------------ | --------- | ----- | ----- | ----- | ----- |
| **Diffusion-TracIn** | 0.807   |       | 0.838 | 0.842 | 0.843 | 0.842 |
| **Diffusion-ReTrac** | 0.940   |       | 0.949 | 0.950 | 0.949 | 0.950 |

Table 4: **Performance on Outlier Model.** Accuracy of Diffusion-TracIn and ReTrac on 16 CIFAR-plane test samples, varying the number of expectation timesteps sampled.

We then evaluate the accuracy drop on the CIFAR-MNIST outlier model as in section 6.1, by accessing the number of CIFAR planes ranked among the top 200 influential when attributing a plane test sample. We observe that compared to full expectation, the performance drop is negligible when averaging more than 50 timesteps. Similarly, a notable decline in accuracy is observed when employing only one timestep (either fixed at 500 or selected randomly) per checkpoint. The result is shown in table 4.

### E.2   Efficiency of Timestep Sparse Sampling

We then evaluate the efficiency improvement of the timesteps subsampling approach compared to the full expectation. Utilizing a subsample of 50 evenly spaced timesteps reduces the computational cost by approximately 20-fold, compared to the full expectation operation. Overall, the findings show that our proposed sparse sampling approach significantly reduces computational time while maintaining high fidelity attribution results, offering a compelling alternative to full expectation methods.

| Time (In seconds)  | 50    | 100   | 200    | Full   |
| ------------------ | ----- | ----- | ------ | ------ |
| Diffusion-TracIn   | 27.55 | 57.67 | 100.01 | 509.41 |
| Diffusion-ReTrac   | 28.87 | 53.39 | 105.09 | 508.97 |

Table 5: **Computational Time.** Measurement of computational efficiency for Diffusion-TracIn and Diffusion-ReTrac, utilizing $n$ evenly spaced timesteps and full (1000) timesteps.

## F  Implementation Details

### F.1  Model Details

The diffusion models utilized in the experiments are trained with Diffusion Denoising Implicit Model (DDIM) (Song et al., 2020) architecture, using 1,000 denoising timesteps and 50 inference steps. Stochastic gradient descent (SGD) and Adam optimizers are used and tested for the attribution results. Additionally, training and experimentations are conducted on Nvidia A10G and RTX A6000 GPUs.

We note that our approach should remain consistent across variations of diffusion models, since the methods are designed based on the training process which is largely unaffected by differences in inference procedures. Furthermore, our methods may also be modified and derived to accommodate the variations in training, such as the optimizer used. Similar to the original work presented in TracIn, the practical form of the attribution method is expected to remain the same across these variations (Pruthi et al., 2020).

### F.2  Checkpoint Selection

Influence estimation is susceptible to the choice of model checkpoints, particularly because model behavior can vary significantly over the training period. It is optimal to select checkpoints with consistent learning and a steady decline in loss, which indicate reliable learning stages. Checkpoints that are too early in the model's learning stage can exhibit fluctuating gradient information due to initial instability, while checkpoints near model convergence offer limited insights into the attribution. Influence estimation at these early or late epochs of the learning process can introduce noise and compromise the accuracy of attribution results.

Attribution methods that rely on loss gradient norm information are also particularly sensitive to checkpoint selection. We observe that certain samples may exhibit an unusually large norm at specific checkpoints. When these checkpoints are used in Diffusion-TracIn, such samples emerge as generally influential with notably high influence on various test samples, overshadowing attribution results from previous checkpoints. This effect is mitigated in Diffusion-ReTrac due to re-normalization, reducing the method's susceptibility to dominating loss gradient norms.

