# OpenReview forum: "Data Attribution for Diffusion Models: Timestep-induced Bias in Influence Estimation"
_TMLR — Accepted by TMLR_

### Review · Reviewer_D4pE · 2024-03-18

**Summary Of Contributions:**

This paper presents two attribution scores to estimate the influence of training samples on the loss of testing samples in diffusion models. The paper proposed to use expectation (Diffusion-TracIn) and gradient norm normalization (Diffusion-ReTrac) to address the timestep-induced norm bias observed when applying conventional data attribution score (TracIn) to the diffusion loss.

**Audience:**

Yes

**Claims And Evidence:**

Yes

**Requested Changes:**

1. Since the timestep sparse sampling is an efficient approximation method, it might lead to reduced or less stable performance. I'm curious about the performance of omitting it from Diffusion-ReTrac. Additionally, I'm curious about how much quicker it is relative to Diffusion-TracIn.
2. The TracIn with different fixed timestep would be a valid baseline in experiments.
3. In section 4.2, before the itemized list, in the sentence “the presence of such timestep-induced norm by showing” should be “timestep-induced norm bias”
4. In equation (4), the definition of the summation range $k:z_k=z$ should be clarified.

**Strengths And Weaknesses:**

Strengths
1. The idea of generalizing data attribution methods to diffusion models for image source tracing is interesting.
2. The observation of the time-induced gradient norm bias is insightful for other research, e.g, improving training stability.

Questions
1. In equation (4), the definition of the summation range $k:z_k=z$ is not clear to me. Is it a condition to select checkpoints $w_k$? I wonder why not every checkpoint could be used in computing the latter gradient term in the summation? Is that some checkpoint has never been updated by the gradient computed with the training sample $z$?
2. Since the timestep sparse sampling is an efficient approximation technique, it might lead to reduced or less consistent performance. I'm curious about how much quicker it is relative to Diffusion-TracIn. Additionally, I'd like to know the performance impact of omitting it from Diffusion-ReTrac.

---

> ### Author Response · Authors · 2024-03-19
> **Response to Reviewer D4pE (Part I)**
>
> Thank you so much for your valuable feedback! We are glad you find our work generalizing data attribution methods to diffusion models interesting, and observation of norm bias insightful for the research community. We address your questions below.
>
> > ### **Q1. Definition of $k: z_k = z$ in eq(4) & Checkpoints**
>
> Thank you for identifying the ambiguity in the notation. Equation 4 is an idealized notion of the approximated influence, which sums over all the iterations $k$, whenever the particular training sample $z$ is utilized. However, since this requires access to model weight at every iteration (or epoch), the computational costs are large. Therefore, in practice, we utilize checkpoints to approximate this training process (Equation 7). We shall modify Section 3.2 to clarify our notations.
>
> > ### **Q2. Timestep sparse sampling**
>
> Thank you for your suggestions. We will add extra experiments to compare 1). performance and 2). computational time of our approximation to the original method. We will report the results as soon as possible and include this comparison in the paper. We will also report more investigation regarding timesteps as mentioned in Requested Changes (4).
>
>
> Thank you for pointing out the additional possible clarifications, we will address these points as suggested.

---

> ### Author Response · Authors · 2024-03-28
> **Response to Reviewer D4pE (Part II)**
>
> We thank the reviewer for the valuable feedback. In response to your suggestions, we have fixed various typos in the paper and added experiments to demonstrate the effectiveness of our timesteps sparse sampling approach in appendix G. A more detailed description of the experiments we conducted is listed below.
>
> > ### **Q1: Performance comparison: timestep sparse sampling vs. full expectation**
>
> A1: We evaluate the effectiveness of timesteps sparse sampling from two perspectives.
> 1. **Consistency**: We compare the Diffusion-TracIn and ReTrac results obtained using our sparse sampling method against those obtained through the full expectation of all timesteps $1\sim1000$. Specifically, we evaluate the Spearman's Rank Correlation between the attribution results derived from the two approaches. To ensure the reliability of our findings, we report the average correlation over $16$ test samples. It is observed that Spearman's rank correlation consistently exceeds $0.99$ across various sparsity levels (i.e., using $50/100/200$ evenly spaced timesteps), for both TracIn and ReTrac. This suggests that our approach, leveraging sparse sampling, yields attribution results that closely align with those derived from full expectation. It is also worth noticing that fixing one timesteps per checkpoint is shown to be unfaithful to the true attribution result, where the correlation drops to 0.73 for Diffusion-ReTrac and 0.72 for Diffusion-TracIn. The correlation scores are organized in the table below:
>
> | **Correlation** | **Fixed one** | **50** | **100** | **200** |
> |-----------------------------|----------------------|--------------|---------|---------|
> | **Diffusion-Tracln**        | 0.720               | 0.995       | 0.998   | 0.999   |
> | **Diffusion-ReTrac**        | 0.730               | 0.995        | 0.997   | 0.999   |
>
> 2. **Performance**: We evaluate the accuracy drop on the CIFAR-MNIST outlier model (section 6.1 Image Tracing), by accessing the number of CIFAR planes ranked among the top $200$ influential when attributing a plane test sample. We observe that compared to full expectation, the performance drop is negligible when averaging more than $50$ timesteps. Similarly, a notable decline in accuracy is observed when employing only one timestep (either fixed at $500$ or selected randomly) per checkpoint.
>
> | **Accuracy** | **Fixed one** | **50** | **100** | **200** | **Full** |
> |-----------------------------|----------------------|--------------|---------|---------|----------|
> | **Diffusion-Tracln**        | 0.807                | 0.838        | 0.842   | 0.843   | 0.842    |
> | **Diffusion-ReTrac**        | 0.940                | 0.949        | 0.950   | 0.949   | 0.950    |
>
>
> > ### **Q2: Efficiency comparison with various number of sampled timesteps**
>
> A2: In response to the comment, we empirically evaluate the computational time comparison with varying sizes of expectation timesteps ($50/100/200$ evenly spaced, and full $1\sim1000$). We organize the empirical time for this expectation operation for 1 test sample in the following table:
>
> | **Time (In seconds)** | **50** | **100** | **200** | **Full** |
> |-----------------------|--------|---------|---------|----------------|
> | **Diffusion-Tracln**            | 27.55  | 57.67   | 100.01  | 509.41         |
> | **Diffusion-ReTrac**            | 28.87  | 53.39   | 105.09  | 508.97         |
>
> Overall, the findings show that our proposed sparse sampling approach significantly reduces computational time while maintaining high fidelity attribution results, offering a compelling alternative to full expectation methods.

---

### Review · Reviewer_Hnoh · 2024-04-07

**Summary Of Contributions:**

The authors study the problem of data attribution in diffusion models. The main contribution of this paper is the experimental finding that the time-dependent training of diffusion models leads to increased loss gradient norms for training examples that were seen under high noise. This norm bias throttles the performance of the standard extension of the TracIn framework to the diffusion setting. To mitigate this, the authors propose a simple modification of the TracIn extension to the diffusion setting that re-normalizes the norms taking into account the training timesteps at which the examples appeared. This seems to lead to increased experimental performance.

**Audience:**

Yes

**Broader Impact Concerns:**

I do not have any broader impact concerns.

**Claims And Evidence:**

No

**Requested Changes:**

I think it would strengthen significantly the submission if the authors:

* Apply their method to more meaningful datasets, such as LAION.
* Show data attribution for open-source, pre-trained models such as Stable Diffusion (XL).
* Include more baselines.

I would be further curious to see:

* How this method performs for diffusion models that are trained with corrupted data, as in Ambient Diffusion. Training under corruption has been proposed as a way to mitigate memorization of training examples. It would be interesting to see whether the proposed attribution method works in this case and whether the phenomenon of increased gradient norms carries to this setting.
* if the method works for latent diffusion models.

**Strengths And Weaknesses:**

Strengths:

* The attribution problem in the context of generative models is very timely and I believe it will be in the interest of the TMLR audience.
* The experimental finding of the correlation between the norm of the loss gradient and the timestep at which a point appeared during training is interesting.
* The authors do a great job in validating this experimental finding through a series of ablations.
* The proposed objective is reasonable and seems to perform very well in the settings in which it was evaluated.

Weaknesses:

* The experimental evaluation is kind of limited. The authors only study toy datasets. This type of work would have a lot more impact if more realistic datasets were considered, such as LAION.
* The method assumes not only access to the training dataset but also access to the training timesteps that were used for each example. This seems kind of limited.
* The authors only compare to one baseline. Further, this baseline was proposed by the authors themselves. I am not very familiar with the literature, but aren't there more works to compare to? Or other natural baselines to consider?
* I did not find any theoretical justification for the experimental finding regarding the correlation between the loss gradient norms and the training timesteps. Is this experimental finding universal? Where is it coming from?

---

> ### Author Response · Authors · 2024-05-03
> **Response to Reviewer Hnoh (Part I)**
>
> We thank the reviewer for the valuable feedback. We are glad that you find our work to be timely, interesting, and well-validated through a series of abortions. In response to your suggestions, we have added the experiments below.
>
> > ### **Requested Changes 1 & Weakness 1**: *Apply method to more meaningful datasets, such as LAION.*
>
> Thank you for the suggestion. We agree that utilizing more realistic and complex datasets can further demonstrate the robustness of our findings, but the scale of datasets like LAION-5B or LAION-400M may pose practical challenges within our computational resources available.
>
> However, we have added an evaluation on two datasets with larger size and higher resolution: 1). Tiny ImageNet containing 100,000 images with 64 x 64 resolution, and 2). CelebA containing 100,000 images with 128 x 128 resolution. Since these datasets are also frequently used in the research community for generative models, we hope these additions contribute to a more comprehensive assessment of our method.
>
> Comparison of Diffusion-TracIn and ReTrac are shown below, using the same metric as Section 6.2 where a lower score indicates more generally influential samples. The results align with the main text, where ReTrac consistently outperforms TracIn by a substantial margin. Further visualization is also available in Appendix E.
>
> |      |   | Top 10 | Top 50 | Top 100 |
> |--------------|----------|--------|--------|---------|
> | Tiny ImageNet| D-TracIn  | 0.586  | 0.474  | 0.439   |
> |              | D-ReTrac  | **0.900**  | **0.841**  | **0.823**   |
> | CelebA       | D-TracIn  | 0.313  | 0.273  | 0.269   |
> |              | D-ReTrac  | **0.956**  | **0.903**  | **0.876**   |
>
>
> > ### **Requested Changes 2**: *Show data attribution for open-source, pre-trained models such as Stable Diffusion (XL).*
>
> Thank you for suggesting the application of our method on open-source, pre-trained models. However, our approach requires access to model checkpoints across various epochs, particularly in the earlier stages where significant model learning occurs. Unfortunately, such detailed checkpoints for early epochs might not be readily available due to the size of pretrained diffusion models, and it's challenging to retrain our own model at a similar scale. We would also highly appreciate the opportunity for those with access to relevant checkpoints to utilize our method.
>
> Additionally, the model utilized in our study adapts the same or similar features found in typical diffusion models. Thus, we believe that our findings would generalize to other diffusion models in use. The specific model architecture employed is also included in our codes for researchers to access and reproduce our findings effectively.
>
>
> > ### **Requested Changes 3 & Weakness 3**: *Include more baselines.*
>
> Thank you for the feedback regarding the additional baselines. In Appendix C, we have accordingly added a performance comparison to influence functions, one of the best-known influence estimators. The same metric of precision curve used in Section 6.1 and example visualizations are available in the appendix.
>
> We note numerous cases where CIFAR-planes are attributed to MNIST-zeros, and vice versa. The performance of influence functions drops especially for attributing MNIST-zero test samples, resulting in a precision consistently lower than 0.4 while ReTrac reaches close to 1.0. Overall, Diffusion-ReTrac outperforms influence functions significantly.
>
>
>
> > ### **Requested Changes 4 & 5**: *Ambient Diffusion & Latent Diffusion models*
>
> Thank you for your insightful feedback regarding the potential application of our method on diffusion models trained with corrupted data, such as Ambient Diffusion. We appreciate your suggestion to investigate this aspect further.
>
> Indeed, exploring how our proposed attribution method performs in scenarios involving corrupted data and latent diffusion models is an intriguing avenue for future research. We are keen to see the attribution results extended to these scenarios. We have added this valuable point to the Conclusion section outlining this potential direction for exploration, including the exploration of our method in the context of corrupted data and latent diffusion models. Thank you once again for highlighting this aspect!

---

> > ### Author Response · Authors · 2024-05-03
> > **Response to Reviewer Hnoh (Part II)**
> >
> > > ### **Weakness 2**: *Access to training timesteps seems limiting.*
> >
> > Thank you for the feedback regarding the assumption of access to the training timesteps in our method. Indeed, we agree that this requirement poses a limitation, and we have promptly added this acknowledgment in our Limitations section.
> >
> > This assumption is important to the theoretical framework underlying Diffusion-TracIn and Diffusion-ReTrac. Unlike in traditional classification settings where having access to training data alone allows the reconstruction of the training process, in the context of diffusion models, the additional temporal trajectory requires knowledge of timesteps to faithfully replay the training process. And thus we have prioritized maintaining fidelity to the theoretical framework in our proposed methods and following evaluations. Thank you for pointing out this aspect, and we appreciate your understanding of the rationale behind this requirement.
> >
> >
> > > ### **Weakness 4**: *Theoretical justification for correlation between norms and timesteps: Is this finding universal? Where is it coming from?*
> >
> > Thank you for the question regarding further explanation of the correlation between norms and timesteps. We acknowledge the absence of a theoretical justification in our paper at present and noted it accordingly in the Limitations section.
> >
> > Our hypothesis regarding this trend is primarily rooted in the learning dynamics inherent to diffusion models. Specifically, previous works such as *[1]* have suggested that later timesteps in diffusion models tend to learn structure, while earlier timesteps focus on capturing finer details. We speculate that this observed correlation between gradient norms and training timesteps may stem from similar dynamics, as evidenced by the variability in regions of high-norm induction across different stages of learning.
> >
> > To demonstrate that this finding is universal, we included in Appendix A.2 additional norm distributions on the following diffusion models: 1). trained on Tiny-ImageNet, 2). trained on CelebA, 3). open-source Google's Pretrained DDPM on CIFAR-10.  Aligning with the results shown in the main text, there is a notable region of timesteps with significantly larger norms.
> >
> > By identifying a critical and unexpected effect due to diffusion timesteps, our research presents an initial investigation into this issue, with potential implications for improving methods and explaining challenges currently encountered by researchers. We envision our work as offering a potential future direction of exploration, contributing to a more comprehensive understanding of diffusion models.
> >
> > **[1]** Fang, Gongfan, Xinyin Ma, and Xinchao Wang. "Structural pruning for diffusion models." Advances in neural information processing systems 36 (2024).

---

### Review · Reviewer_WxYC · 2024-04-22

**Summary Of Contributions:**

This work proposes a new data attribution method Diffusion-ReTrac for the diffusion model, using a similar manner of TracIn but considering the influence estimation bias from different timesteps. The key observation is that the magnitude of the loss gradient norm is highly correlated with the timestep of the sample. While naive TracIn extension Diffusion-TracIn only includes the expectation over timesteps and noise $\epsilon$ with the existence of diffusion temporal bias,  Diffusion-ReTrac addresses the dominating gradient norm effect by additional introducing normalization for gradient terms. The paper provides experimental results demonstrating the effectiveness of Diffusion-ReTrac on the diffusion model compared with TracIn naive extension.

**Audience:**

Yes

**Broader Impact Concerns:**

No broader impact concerns.

**Claims And Evidence:**

Yes

**Requested Changes:**

1. More experiments for answering questions mentioned in weakness part will be great, e.g. show whether the influence ranking of a sample is consistent through timesteps (Diffusion-ReTrac only includes 1 timestep) with the normalization manner of Diffusion-ReTrac.
2. Additional experiments for more potential baselines.

**Strengths And Weaknesses:**

Strengths:
1. Well extension of TracIn for the diffusion model considering the training temporal dynamic of diffusion.
2. This work provides comprehensive experiments to show the strong correlation between the timesteps and the magnitude of the gradient norm, indicating the necessity of an influence estimation method resilient to timestep-induced gradient norm bias for the diffusion model.
3. Well written and interesting.

Weaknesses:
1. The proposed Diffusion-ReTrac implicitly assumes that influence estimation from each timestep contributes equally as the normalized gradients share the same gradient norm. However, for each training sample, its influence on the test sample may vary through each timestep. The gradient norm difference might be from 2 things: (1)  gradient norm bias mentioned in the paper and (2) samples contribute differently during training at different timesteps. Could you provide more evidence to support your choice?
2. The outlier detection result that Diffusion-ReTrac performs worse than Diffusion-TracIn comes from the same reason mentioned in weakness 1.  The role of critical timestep with abnormal self influence value is weakened. However, if considering the more precise norm of each timestep (instead of the same norm),  this drawback above might be alleviated.
3.  Lack of comparison between Diffusion-ReTrac and related methods, e.g. similar influence re-normalization/reweight manner.

---

> ### Author Response · Authors · 2024-05-03
> **Response to Reviewer WxYC (Part I)**
>
> We thank the reviewer for the valuable feedback. We are glad that you find our work as a well extension of TracIn, provides comprehensive experiments highlighting the necessity of addressing bias, well written and interesting. We address your questions below.
>
> > ### **Weakness 1**: *Each training sample's influence may vary on different timesteps. Thus, the gradient norm difference might also come from samples contributing differently at their training timesteps. Could you provide more evidence to support your choice?*
>
> Thank you for highlighting this perspective. We acknowledge the reviewer's concern regarding the varying influence of each training sample across different timesteps. This is indeed an aspect that our proposed method may overlook, and it's also something we have carefully considered when choosing our current approach Retrac. We hope to share our thoughts behind this choice below:
>
> It's true that treating every timestep equally may obscure the significance of certain critical timesteps that better demonstrate the influence of training samples. However, we opted for this normalization approach to counteract the overly dominant influence observed in the vanilla approach Diffusion-TracIn. In Diffusion-TracIn, if a training sample $x$ has a timestep that falls in a large-norm inducing region (such as timesteps close to 1000 in Figure 1), then this sample $x$ often emerges as one of the top proponents for most test samples in calculation. We observe that this could be the case even when $x$ only has a large-norm timestep in one checkpoint, which means that this effect is strong enough to overshadow the attribution results from all previous checkpoints.
>
> This tendency is undesirable because it overly emphasizes certain training samples even if their influence is not consistently significant across all checkpoints, obscuring the nuanced dynamics of the model's learning process. By preventing this over-dominance, Diffusion-ReTrac aims to provide detailed and specific insights into the contributions of individual training samples. This is notable as the "generally influential" sample is no longer an issue in ReTrac attribution results.

---

> ### Author Response · Authors · 2024-05-03
> **Response to Reviewer WxYC (Part II)**
>
> > ### **Weakness 2 & Requested Changes 1**: *How does the intensity of normalization affect attribution results? Is there a potential approach that offers a more precise handling of each individual training sample?*
>
> Thank you for raising the insightful query. We present our thoughts as follows:
>
> In Figure 3, we have shown that for the same training sample, if we plot its loss gradient norm distribution over every single timestep, we also observe that there is a region of gradual increase and peak. This trend is also consistent for other random samples analyzed, where norm is found to be highly dependent on timesteps. Thus we argue the norm of a training sample is comprised of two major components: (1). natural norm that is fully dependent on the training sample's properties, and (2). dynamic norm that is highly dependent on timestep; and norm 2 is what we currently consider to be biasing the attribution.
>
> As the reviewer points out, while ReTrac mitigates the overly-dominant effect of norm 2, it also diminishes norm 1 which may be helpful towards attribution. Here we include one prior attempt to address the imprecise handling of norms by utilizing ***"guided" normalization***, which aims to only remove timestep-induced dynamic norms, while maintaining the natural norm informative for attribution. For an overview, we fix a timestep $t$ (e.g. the highest norm-inducing timestep at that checkpoint), and compute the norm for every training sample at $t$. Since the effect of timestep is now controlled for all samples, this resulting variation in norms is sample-induced. Hence we utilize this variance between samples to guide us in answering the question *how much to normalize for each sample?*
>
> A new section with details and visualization comparing the methods ReTrac, Guided, and TracIn is shown in Figure 11 of Appendix B. There is indeed a trend showcasing the inherent trade-off associated with the three methods: a higher degree of normalization reduces the presence of generally influential proponents in attribution results and mitigates the influence of large-norm MNIST samples, at the cost of slightly diminished performance in attributing MNIST-zero samples.
>
> Nonetheless, since we did not observe an immensely convincing improvement with guided normalization, we ultimately opted for the ReTrac normalization approach as a simpler and straightforward approach to address the issue. Our experiment results (e.g. Section 6.1 - 6.2)  have also shown that the increase in performance is significantly greater than the drawback of ~3% in outlier detection (Section 6.3), deeming ReTrac to be a more desirable choice providing targeted attribution. However, further refinement of Guided Normalization and other strategies for optimal normalization may help pinpoint the optimal balance. We have also noted this as a potential direction to explore in Appendix B.
>
> We sincerely thank the reviewer for highlighting this aspect. We hope that this provides clarification to support our methodological choices.
>
>
> > ### **Weakness 3 & Requested Changes 2**: *Additional experiments for more potential baselines*
>
> Thank you for the feedback regarding the additional baselines. In Appendix C, we have accordingly added a performance comparison to influence functions, one of the best-known influence estimators. The same metric of precision curve used in Section 6.1 and example visualizations are available in the appendix.
>
> We note numerous cases where CIFAR-planes are attributed to MNIST-zeros, and vice versa. The performance of influence functions drops especially for attributing MNIST-zero test samples, resulting in a precision consistently lower than 0.4 while ReTrac reaches close to 1.0. Overall, Diffusion-ReTrac outperforms influence functions significantly.

---

### Author Response · Authors · 2024-05-24
**Official Comment by Authors**

We thank the reviewers for their insightful feedback and suggestions. We believe we have addressed all the questions and suggested changes raised, and have updated the paper based on the reviews (changes are highlighted in blue). We appreciate the reviewers' assistance in improving our paper!

___
> ### Here we summarize the major changes in our paper since initial submission:

1. Attribution results on two additional datasets (Appendix E)
2. Baseline: different normalization techniques (Appendix B)
3. Baseline: influence functions (Appendix C)
4. Further analysis into the effectiveness and efficiency of timestep sparse sampling to approximate the entire training trajectory (Appendix G)
5. Loss gradient norm distributions on additional models, including open-source pretrained models (Appendix A.2)

___
> ### We have addressed the remaining points raised in the responses:

1. Further motivation for normalization in Diffusion-ReTrac as opposed to TracIn, including the potential trade-offs involved
2. Preliminary intuition for the correlation between loss gradient norms and timesteps, along with further evidence that this finding is universal in the diffusion models examined
3. Explanation for the required access to training timesteps

We thank the reviewers once again for their time and valuable insights. If there is anything else that we can discuss or clarify, we are happy to provide more information as needed.

---

### Decision · Action_Editor_vwLu · 2024-06-03

**Recommendation:** Accept with minor revision

**Comment:**

This paper tries to develop data attribution methods to interpret diffusion models. Different from traditional models, diffusion models operate over a sequence of timesteps instead of instantaneous input-output relationships as in previous contexts. This paper proposes Diffusion-TracIn to incorporate this temporal dynamics, and further proposes Diffusion-ReTra to overcome limitations of Diffusion-TracIn.

The reviewers mostly agree on the value of the proposed problem and solutions. They raised some questions mainly on experiments, limitations and some technical details. The rebuttal address all the problems, and the reviewers all feel positive of the paper after additional experiments and explanations in the rebuttal. I think the paper is a first exploration of data attribution in diffusion models, and is worth of publishing.

I request the authors to fully address the reviewer comments into the final revision. I see the authors have added multiple changes and extra experiments in the rebuttal, I would suggest the authors at least to mention them in the main paper and put details in the appendix.

**Audience:**

Researchers in data attribution method and diffusion models might find the work interesting and valuable.

**Claims And Evidence:**

Yes, the claims are generally supported by convincing and clear evidence, through both experiments and discussions.

---

> ### Author Response · Authors · 2024-07-05
> **Response by Authors**
>
> Thank you for your comprehensive evaluation and the final decision!
>
> We agree and are committed to integrating all suggested changes and clarifications raised during the review process. In response to the valuable feedback, the camera-ready manuscript now includes the additional experiments and details that address all comments from the rebuttal.
>
> We are grateful for your recognition of our work's potential impact. Thank you again for your time and effort.